# Direction-dependent dynamics of colloidal particle pairs and the Stokes-Einstein relation in quasi-two-dimensional fluids

Noman Hanif Barbhuiya[1], A. G. Yodh [2] & Chandan K. Mishra [1]✉

Hydrodynamic interactions are important for diverse fluids, especially those with low Reynolds number such as microbial and particle-laden suspensions, and proteins diffusing in membranes. Unfortunately, while far-field (asymptotic) hydrodynamic interactions are fully understood in two- and three-dimensions, near-field interactions are not, and thus our understanding of motions in dense fluid suspensions is still lacking. In this contribution, we experimentally explore the hydrodynamic correlations between particles in quasi-two-dimensional colloidal fluids in the near-field. Surprisingly, the measured displacement and relaxation of particle pairs in the body frame exhibit direction-dependent dynamics that can be connected quantitatively to the measured near-field hydrodynamic interactions. These findings, in turn, suggest a mechanism for how and when hydrodynamics can lead to a breakdown of the ubiquitous Stokes-Einstein relation (SER). We observe this breakdown, and we show that the direction-dependent breakdown of the SER is ameliorated along directions where hydrodynamic correlations are smallest. In total, the work uncovers significant ramifications of near-field hydrodynamics on transport and dynamic restructuring of fluids in two-dimensions.

The investigation of hydrodynamics in fluids at low Reynolds numbers has a venerable history and continues to yield surprises[1–16]. Generally, particle transport in such fluids is influenced by hydrodynamic interactions, which span near- and far-field length scales[4–6], and which depend strongly on spatial confinement (dimension) and fluid boundary conditions[17,18]. In three dimensions (3D), monopole-like hydrodynamic interactions give rise to drag forces on particles in particle-pairs of the same sign in both longitudinal ($L$) and transverse ($T$) directions[10,17]. The body-frame ($L, T$) − axes are along and perpendicular to the line joining the particle-pair at the initial time $t_0$ (Fig. 1a). By contrast, in two-dimensions (2D), the asymptotic far-field hydrodynamic solutions exhibit a dipolar flow profile with longitudinal drag and transverse antidrag coupling between particles in particle-pairs[4,6]. In addition, as the particle packing fraction increases, while analytic solutions to the Stokes flow are challenging to obtain, experiments

have shown that near-field drag correlations exhibit oscillatory modulations with respect to particle separation that are in-phase with structural signatures such as the particle pair correlation function[19]. The nature of transverse antidrag coupling in the near-field, however, deviates from the far-field dipolar flow profile and remains largely unexplored; for example, phase differences between transverse and longitudinal correlations could exist and, if so, could have consequences in dense suspensions and confined geometries.

Our experiments on confined colloidal suspensions in 2D shed light on these issues, revealing direction-dependent transport of colloids in the body-frame of colloid pairs caused by the contrast in strength between longitudinal drag and transverse antidrag, as well as the phase difference between them. The influence of such anisotropic hydrodynamic interactions on the Stokes-Einstein relation (SER) for fluids in confined geometries (or otherwise) is unexplored. The SER

[1]Department of Physics, Indian Institute of Technology Gandhinagar, Palaj, Gandhinagar 382055 Gujarat, India. [2]Department of Physics and Astronomy, University of Pennsylvania, Philadelphia 19104 PA, USA. ✉e-mail: chandan.mishra@iitgn.ac.in

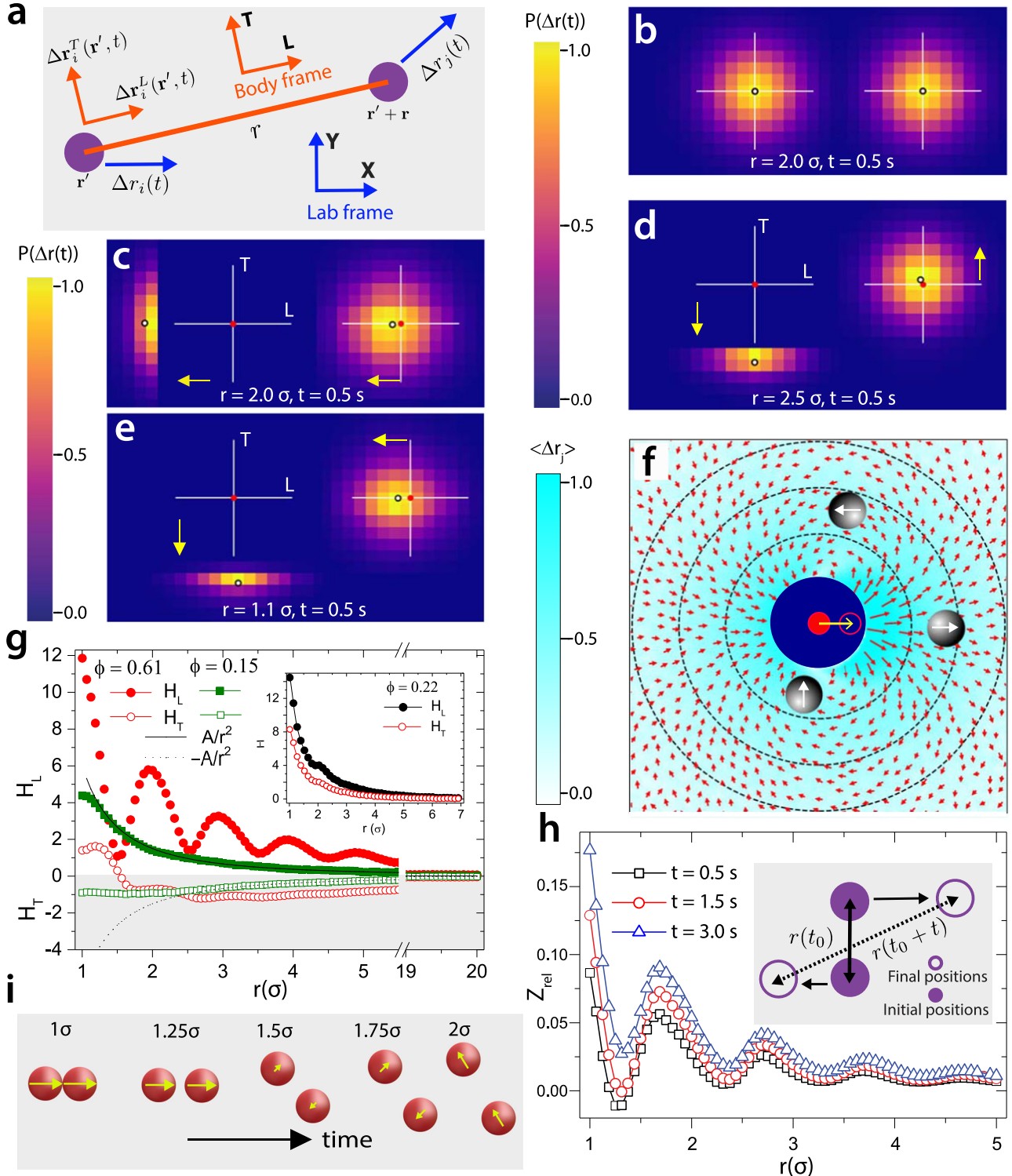

**Fig. 1 | Visualizing hydrodynamic modes and spatiotemporal evolution of pairs. a**, Schematic showing the body-frame, $(L, T)$ − axes, for a colloid-pair used to reveal hydrodynamic correlations. **b**, Colormap of $P(\Delta \mathbf{r}_i(\mathbf{r}', t)|\mathbf{r}_j = \mathbf{r}' + \mathbf{r})$ and $P(\Delta \mathbf{r}_j(\mathbf{r}' + \mathbf{r}, t)|\mathbf{r}_i = \mathbf{r}')$ for particles in pairs separated by $r = 2.0\sigma$. Conditional $P(\Delta \mathbf{r}(t))$ measured for the particle on the right of the pair when the particle on the left displaces by $\geq 1.0\sigma$ **c**, along $L$ and $r = 2.0\sigma$ **d**, along $T$ and $r = 2.5\sigma$ and **e**, along $T$ and $r = 1.1\sigma$, depicted by yellow arrows. Solid red and white circles represent the mean positions of the particles at $t_0$ and $(t_0 + t)$, respectively. The displacement color map for **b−e** are normalized by the maximum displacements in each case. **f**, Polar colormap, $\mathbf{r}(r, \theta)$, of displacement field when the colloid at origin (solid red

circle) moves towards the right (open red circle). The dashed radial circles represent $r = \{2, 3, 4\}\sigma$. Representative red arrows, with their head and length represent the direction and strength (shown also in the background), respectively, of the field. The measurements for **b−f** were performed at $\phi = 0.15$ and $t = 0.5$ s. **g**, $H_L$ and $H_T$ versus $r$ for quasi-2D experimental cell. The inset shows $H_L$ and $H_T$ versus $r$ for open (3D) experimental cell. **h**, $Z_{rel}$ versus $r$ at $\phi = 0.61$. The inset shows typical schematics depicting configurations corresponding to the peak position in $Z_{rel}$. **i**, "Most-probable" schematic construction of spatiotemporal evolution of a pair of particles due to near-field hydrodynamics. Note, the relative magnitudes and directions of the yellow arrows correspond to expectation for each configuration.

explicitly connects the bulk viscosity of fluids to the microscopic self-diffusivity of tracers and is central to diverse processes, such as drug delivery, and physics in emulsions and biological systems[2,20–22]. Interestingly, although originally derived for dilute suspensions, the SER has been found to be valid in both dilute and dense suspensions in three and higher spatial dimensions[23]. However, the microscopic origins of its breakdown in the reduced dimensions remains unclear[23–26]. Our experiments reveal unique features of 2D hydrodynamic interactions that lead to violation of the SER in confined geometries and identify scenarios that recover the SER. This work takes steps towards elucidating the microscopic origins of SER breakdown in reduced dimensions which to date have been unclear[23–26].

## Results

We employ optical video microscopy to probe hydrodynamic interactions of aqueous colloidal suspensions in quasi-2D sample cells (Methods). The experiments were performed with suspensions of micron-size polystyrene latex beads (with diameter, $\sigma$) in their liquid phase as a function of packing area-fraction, $\phi$. Cursory examination of the basic displacement correlation data (Fig. 1) reveals central observations of the experiment. In Fig. 1b, we show the single particle displacement distribution after lag time, $t$, for a pair of particles $\{i, j\}$ located at positions (in 2D) $\{\mathbf{r}', \mathbf{r} + \mathbf{r}'\}$, respectively; the particles are spatially separated by, $\mathbf{r}$, at the initial time, $t_0$ (Fig. 1a). The single particle displacement distributions for each particle are $P(\Delta\mathbf{r}_i(\mathbf{r}', t)|\mathbf{r}_j = \mathbf{r}' + \mathbf{r})$ and $P(\Delta\mathbf{r}_j(\mathbf{r}' + \mathbf{r}, t)|\mathbf{r}_i = \mathbf{r}')$. Here, displacement distributions are obtained from all $t_0$ and for all pairs of particles separated by distance $r$ at $t_0$; $r$ is binned in the intervals of $\sigma/16$. As expected, since particle velocity distributions (and positions) are isotropic in the lab frame of reference, these distributions are measured to be spatially symmetric about their initial position at $t_0$ regardless of $r$ and $\phi$.

The nature and influence of the hydrodynamic interactions between particles in a colloid-pair are revealed in other panels of Fig. 1 (and in other figures) starting with measurements of the conditional probability distribution for particle displacements: $P(\Delta\mathbf{r}_j(\mathbf{r}' + \mathbf{r}, t)|\Delta\mathbf{r}_i^{L, T}(\mathbf{r}', t))$. Here $L$ and $T$ denote the longitudinal and transverse axes of the particle pair in the body frame, oriented, respectively, along and perpendicular to the line joining the particles separated by $\mathbf{r}$ at $t_0$ (Fig. 1a). $P(\Delta\mathbf{r}_j(\mathbf{r}' + \mathbf{r}, t)|\Delta\mathbf{r}_i^{L, T}(\mathbf{r}', t))$ represents the probability that the $j^{th}$ particle will experience displacement $\Delta\mathbf{r}_j(t)$, when the $i^{th}$ particle, separated by $\mathbf{r}$, is displaced by $\Delta\mathbf{r}_i(t)$ along the longitudinal ($L$) or transverse ($T$) direction in the body frame. Figure 1c–e show example data associated with the conditional distribution of particle displacements. Here, the $i^{th}$ particle is displaced by $\geq \sigma$ during a time lag of $t = 0.5$ s, either along $L$ (Fig. 1c) or along $T$ (Fig. 1d & e), and the distribution of the $j^{th}$ particle displacement is shown. These conditional probability distributions provide exemplar exhibits of the well-known hydrodynamic dipolar interaction modes[4,6], showing co-diffusion (drag) of particles in a pair along $L$ (Fig. 1c & Supplementary Video 1) and "anti-symmetric" diffusion (antidrag) of particles in a pair along $T$ (Fig. 1d & Supplementary Video 2).

Notice, in Fig. 1e, the linear superposition of near-field drag and antidrag hydrodynamic fields produces circumferential motion of one particle around the other, thereby leading to a "mass void filling" motion of one particle created by the motion of its partner (Fig. 1e & Supplementary Video 3). To the best of our knowledge, the full character of this type of mass void filling motion (Fig. 1e) has not been directly observed in the experiment. The motional effects caused by near-field drag and antidrag (and their linear combination) are apparent in the polar plot of the hydrodynamic displacement field profile, which we derive using the ensemble-averaged displacement correlations of particles in a pair with separation vector, $\mathbf{r}(r, \theta)$, at $t_0$ and at low $\phi = 0.15$ (Methods, Fig. 1f).

Figure 1 also shows the measured longitudinal and transverse displacement correlation functions, $H_L$ and $H_T$, respectively, versus $r$.

The longitudinal (transverse) hydrodynamic correlation is defined as $H_{L(T)}(r, t) = \langle\Delta\mathbf{r}_i^{L(T)}(\mathbf{r}', t) \cdot \Delta\mathbf{r}_j^{L(T)}(\mathbf{r}' + \mathbf{r}, t)\rangle/D^{self}$ [4,6]. Here, $\Delta\mathbf{r}_i^L$ ($\Delta\mathbf{r}_i^T$) is the displacement in lag time, $t$, of the $i^{th}$ – particle in the $\{i, j\}$ – pair along the $L$ ($T$) direction; the averaging, $\langle\rangle$, is performed over all initial times, $t_0$, and all possible unique pairs $\{i, j\}$. Normalization of $H_{L,T}$ by the $\phi$ – dependent single-particle diffusivity, $D^{self}$, facilitates comparison of $H_{L, T}$ across different $\phi$ (Methods). Figure 1g shows $H_L$ and $H_T$ versus $r$ for two different packing fractions. At low $\phi$ ($\phi = 0.15$), $H_L$ and $H_T$ exhibit expected dipolar decay profiles in the far-field, i.e., they decay as $1/r^2$. A distinguishing feature of quasi-2D fluid confinement is the positive and negative value of the correlation function amplitude of $H_L$ and $H_T$, respectively; when we remove the rigid confining glass wall at the top of the experimental cell and thereby increase sample dimensionality to 3D, the amplitudes of both $H_L$ and $H_T$ are positive (inset to Fig. 1g). At higher density, $\phi = 0.61$, local structural features emerge in the near-field (SI Fig. S1). Specifically, oscillatory spatial modulation of the amplitude appears in both $H_L$ and $H_T$, and the hydrodynamic correlation functions deviate from the dipolar form. Nevertheless, in the far-field ($r > 8\sigma$) even at large packing fraction, the profiles decay as $1/r^2$ (Fig. 1g).

As reported in previous studies, our measurements find that the spatial modulation of $H_L$ (in dense suspensions) is in-phase with oscillation of the colloidal fluids structural pair correlation function (SI Fig. S1)[4,19]. Surprisingly, we find that the antidrag spatial modulations associated with $H_T$ exhibit a spatial phase-shift (phase difference/lag) of around $0.25\sigma$ with respect to $H_L$ (SI Fig. S2). This effect is also revealed by the function $Z_{rel}(r, t) \equiv \langle\frac{r(t + t_0)}{r(t_0)}\rangle_{r', t_0} - 1$ (Fig. 1g). $Z_{rel}(r, t)$ represents the fractional change in the separation of colloids in the colloid-pair during lag time $t$. $Z_{rel}$ clearly captures the antidrag influence on pair-rotation and colloid-pair separation. At the highest $\phi$ ($\phi = 0.61$), $Z_{rel}$ shows oscillatory decaying modulations that are in-phase with the modulations of $H_T$ in the near-field (Fig. 1h).

The insights offered by $H_L$ and $H_T$ (and $Z_{rel}$) suggest a "most-probable" spatiotemporal evolution of particles in a colloid-pair as a function of the particle separation, $r(t)$. This evolution is schematically shown in Fig. 1i. Initially, the particles in the pair are separated by a small distance, $r \sim 1.0\sigma$; they then diffuse and separate to $r \sim 1.25\sigma$. When $r \sim 1.25\sigma$, the pairs are in their most stable state, i.e., they reside in first minima of $Z_{rel}$ (Fig. 1h), and longitudinal drag is dominant. When the separation between particles increases further to $r \sim 1.5\sigma$, then the transverse rotation of the particles begins and leads to further radial separation. When $Z_{rel}$ reaches a maximum at $r \sim 1.75\sigma$, $H_T$ is comparatively stronger and the pair configuration destabilizes. Over time, as $r$ increases, antidrag weakens, and drag becomes dominant again at $r \sim 2.0\sigma$. The cycle will then repeat, but the hydrodynamic interactions become attenuated at larger $r$ (Fig. 1g and h).

The emergent spatiotemporal mobility landscape, revealed by our experiments in the near-field, leads to a particular effective local viscosity and diffusivity associated with the motions of individual particles in the colloid-pair. The experiments thus offer an opportunity to explore the particle-separation-dependent ($r$ – dependent) validity of a physics rule, the Stokes-Einstein relation (SER) in quasi-2D, which has not been explored. Recall, the SER relates the particle diffusion coefficient, $D$, to the viscosity, $\eta$, of the suspending fluid: $D = \frac{k_B T}{6\pi\eta(\sigma/2)}$, where $k_B T$ is the thermal energy. In practice, the structural relaxation time, $\tau_\alpha$, is often used as a proxy for $\eta$[23,27,28]. Simulations of $D$ and $\tau_\alpha$ in 3D liquids demonstrate $D \propto \tau_\alpha^\xi$, with expected SER exponent $\xi = -1$[23]. However, recent computer simulations and experiments in 2D fluids have observed $\xi < -1$[23–25]. The origin of this unusual behaviour, which apparently violates the SER, is unresolved. One interesting suggestion alludes to the presence of long-wavelength Mermin-Wagner fluctuations in 2D liquids[26]. These correlations due to Mermin-Wagner fluctuations can be removed by considering the relative motion of particles with respect to their cages[29,30], which, after implementation, recovers $\xi \sim -1$[26], and thereby suggests that Mermin-Wagner

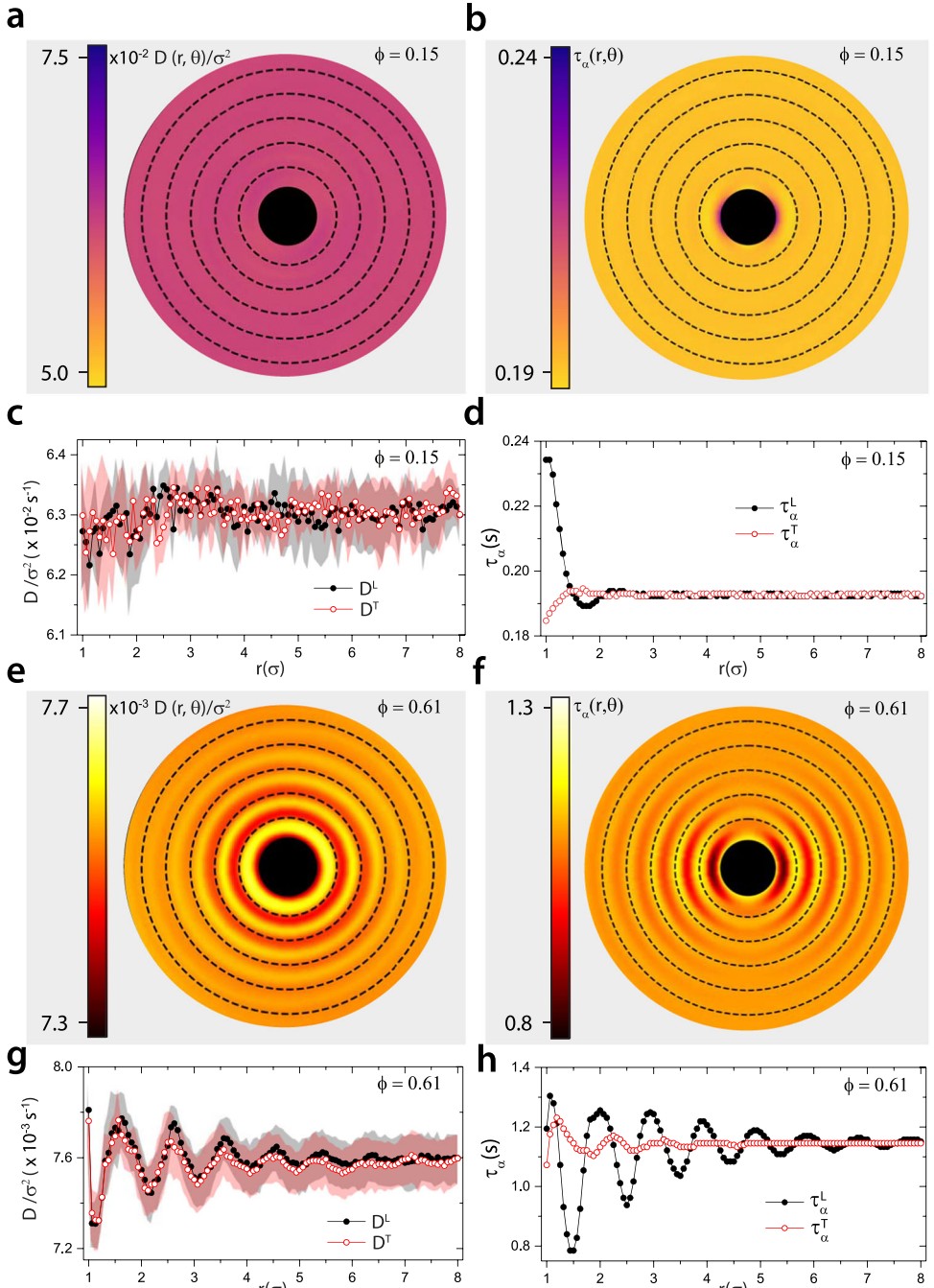

**Fig. 2 | Elucidating the influence of hydrodynamics on transport quantifiers.** Polar colormaps versus $r$ with color-scale on left for $D(r, \theta)$ for **a**, $\phi = 0.15$ and **e**, $\phi = 0.61$, and $\tau_\alpha(r, \theta)$ at **b**, $\phi = 0.15$ and **f**, $\phi = 0.61$. The dashed radial circles are at $r = \{2, 3, 4, \ldots\}\sigma$. Plots along $L$ and $T$ directions corresponding to $\theta = 0°$ and $\theta = 90°$, respectively, for $D(r)$ at **c**, $\phi = 0.15$ and **g**, $\phi = 0.61$, and $\tau_\alpha(r)$ at **d**, $\phi = 0.15$ and **h**, $\phi = 0.61$. The error bars in $D$ are from fittings.

fluctuations cause the anomalous SER exponent. However, strictly speaking, this approach to filter out correlated motions assumes that $D$ and $\tau_\alpha$ are isotropic, *i.e.*, the approach assumes that near-field dynamics have zero angular dependence.

Since we have measured longitudinal and transverse hydrodynamic correlations and particle displacements in the near- and far-field, our experiments offer means to revisit the SER in 2D and to directly investigate the influence of spatial phase differences between longitudinal and transverse hydrodynamic modes. Specifically, we study the separation and angular dependence associated with colloid-pair dynamics based on measurements of single particle diffusion, $D(r, \theta)$, and relaxation, $\tau_\alpha(r, \theta)$ (Methods). Here, $r$ is the particle

separation distance in a pair at $t_0$, and $\theta$ is the angle between the probing direction and the longitudinal axis in the body-frame, $L$ (Methods). At low $\phi$ ($\phi = 0.15$), $D(r, \theta)$ is measured to be isotropic (Fig. 2a & c), but $\tau_\alpha(r, \theta)$ is not; for $r < 2.0\sigma$, $\tau_\alpha(r, \theta)$ is found to be anisotropic (Fig. 2b and d). This $\theta$ − dependence is readily understood. Since drag leads to co-diffusion of particles in colloid-pairs along the longitudinal direction, when $r \sim 1.25\sigma$, $H_T$ is substantially smaller than $H_L$, and particles in the colloid-pairs will take longer to relax along $L$ than along $T$: $\tau_\alpha^L(r \sim 1.25\sigma) > \tau_\alpha^T(r \sim 1.25\sigma)$ (Fig. 2b and d). The oscillatory structural features in the near-field become more pronounced when the particle packing area-fraction is increased. $D(r, \theta) = D(r)$ is still measured to be angularly isotropic and exhibits oscillations as a

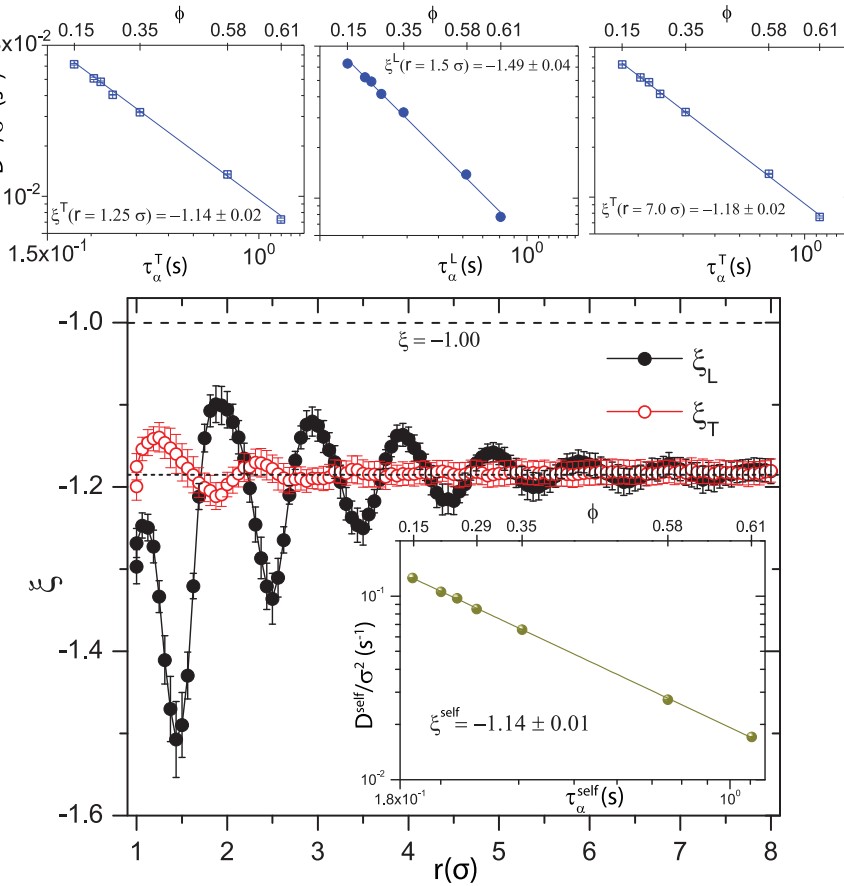

**Fig. 3 | Influence of hydrodynamics on Stokes-Einstein relation in near-field.** SE exponents $\xi^L$ and $\xi^T$ versus $r$. The inset plots the particle self-diffusivity (derived from measurements in the lab frame), $D^{self}$, versus relaxation, $\tau_\alpha^{self}$; the solid line shows $D^{self} \propto \tau_\alpha^{-1.16 \pm 0.03}$. Black dashed and dotted lines at $\xi = -1.00$ and $\xi = -1.18$ depict the ideally expected and measured asymptotic values of $\xi$, respectively. Top panel shows representative $D^L$ and $D^T$ versus $\tau_\alpha^L$ and $\tau_\alpha^T$, respectively, for different $r$ as shown in the figures. The solid lines depict linear fits to determine $\xi$. Standard error from power-law fittings between $D^{L,T}$ and $\tau_\alpha^{L,T}$ are used in $\xi^{L,T}$ versus $r$ plots; systematic errors, obtained by extraction of $D^{L,T}$ from different time-windows, are found to be larger than standard error and are used when quoting the value of $\xi^{L,T}$ in the main text and figure captions.

function of $r$ that are in-phase with $H_L$ (Fig. 2e & g), but $\tau_\alpha(r, \theta)$ is measured to be oscillatory with $r$ and is strongly anisotropic (Fig. 2f and h) due to contrasting magnitude of $H_L$ and $H_T$, and the spatial phase-lag of $\sim 0.25\sigma$ in $H_T$ with respect to $H_L$. This behaviour is readily apparent in Fig. 2d and h, which shows the different variations of $\tau_\alpha^L$ and $\tau_\alpha^T$.

To explore influence of anisotropy in the hydrodynamic correlations on the validity of the SER, we measured $D(r, \theta)$ and $\tau_\alpha(r, \theta)$ for all packing fractions, $\phi$. For simplicity, our discussion will focus on data taken along the longitudinal ($L, \theta = 0°$) and transverse ($T, \theta = 90°$) directions as a function of $\phi$ for fixed $r$. Specifically, for each $r$, we measure the power-law relationship between $D^{L,T}$ and $\tau_\alpha^{L,T}$ using all $\phi$. Exemplar plots and extracted exponents $\xi^{L,T}$ are shown in the top panel of Fig. 3. The resultant variations of $D$ and $\tau_\alpha$ with $r$, and the anisotropy in $\tau_\alpha$ along $L$ and $T$ are reflected in the SER exponents, $\xi^L$ and $\xi^T$, respectively (Fig. 3). Notably, the SER exponents associated with the spatial directions $L$ and $T$ differ from negative unity and differ from each other. Moreover, the spatial phase lag of $\sim 0.25\sigma$ observed for $H_L$ and $H_T$ is also apparent in $\xi^L$ and $\xi^T$. By contrast, if instead we derive $\xi$ from measurements of $D$ and $\tau_\alpha$ along two randomly chosen orthogonal directions in the lab frame (different from $L$ and $T$ in the body-frame), then we find that the $\xi$ are essentially in-phase and identical within experimental certainty (SI Fig. S3). Together, these observations suggest that the unusual trends of $\xi(r, \theta)$ that are apparent in the body frame are due to the distinct motional modes associated with near-field hydrodynamic correlations that arise in 2D colloidal fluids.

The consequences of the near- and far-field hydrodynamic correlations persist in our measurements of the traditional single particle diffusivity, $D^{self}$, and the traditional fluid structural relaxation, $\tau_\alpha^{self}$. The $\phi$ − dependent $D^{self}$ and $\tau_\alpha^{self}$ yield $\xi^{self} = -1.16 \pm 0.03$ (Inset to Fig. 3). Notice, the spatial modulations of the body-frame $\xi$ decay with $r$ and converge to $\xi^{self}$ in the far-field, $r > 8\sigma$ ($\xi^{L,T} \rightarrow -1.18 \pm 0.03$). Evidently, the hydrodynamic interactions in quasi-2D that lead to direction-dependent dynamics and SER violation in the body frame, also lead to violation of the SER in the lab frame.

Finally, we consider whether it might be possible to recover the $\xi \sim -1$ SER exponent for quasi-2D colloidal fluids, perhaps along special directions. To this end, we propose a simple approach again based on the colloid-pairs and their correlated interactions and displacements. Generally, validity of the SER ($\xi \sim -1$) is expected for purely random processes. Since hydrodynamic correlations are smallest along the direction perpendicular to the centre-of-mass displacement ($CM^\perp$) of the colloid-pair (inset to Fig. 4a), one might expect that extraction of $D^{CM^\perp}$ and $\tau_\alpha^{CM^\perp}$ along this direction could yield $\xi^{CM^\perp} \sim -1$. The data in Fig. 4a corroborates these hypotheses. In the far-field, $r > 8\sigma$, where the spatial modulations in $H_L$ and $H_T$ are diminished (Fig. 1f), we find that $\xi^{CM^\perp}$ decays and saturates to $-1.01 \pm 0.02$ (Fig. 4a). In the near field, $r < 8\sigma$, $\xi^{CM^\perp}$ oscillates around negative unity. Note, strictly speaking, the validity of the SER is expected to hold only in the asymptotic limit ($r \rightarrow \infty$), since the solutions that yield the Stokes relation are met only for dilute suspensions and in the asymptotic regime[10]. Interestingly, in this regime ($r < 8\sigma$), $\xi^{CM^\perp} \rightarrow -1$ at specific $r$ that correspond to the

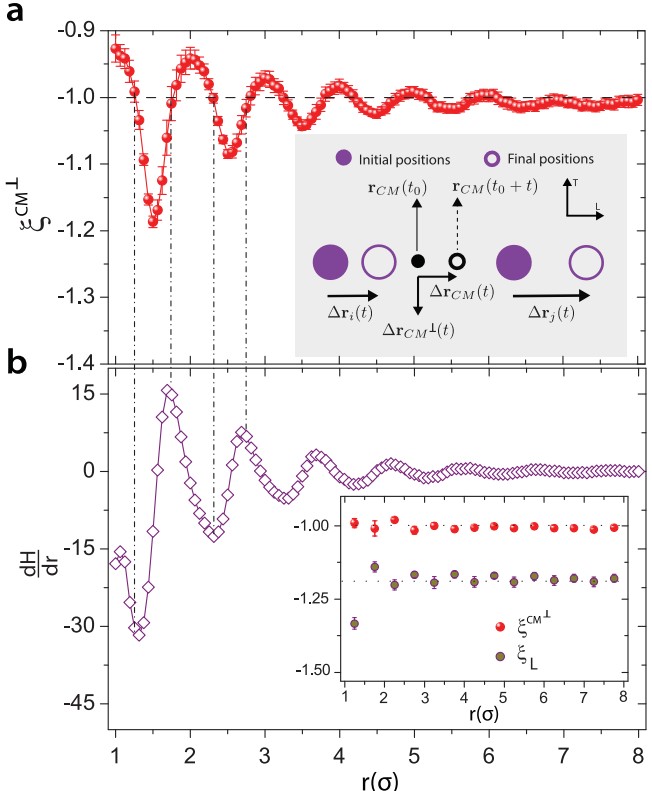

**Fig. 4 | Recovery of the expected Stokes-Einstein exponent. a,** $\xi^{CM\perp}$ versus $r$. $\xi^{CM\perp} \to -1.01 \pm 0.02$ for $r > 8\sigma$; the ideal expected value of $\xi$ is shown by black dashed line. Standard error from power-law fittings between $D^{CM\perp}$ and $\tau_\alpha^{CM\perp}$ are used in $\xi^{CM\perp}$ versus $r$ plot; systematic errors, obtained by extraction of $D^{CM\perp}$ from different time-windows, are found to be larger than standard error and are used when quoting the value of $\xi^{CM\perp}$ in the main text and figure captions. Inset: schematic to visualize the direction in which the displacements of particles in the pair are least correlated, i.e., direction perpendicular to the centre-of-mass displacement direction. **b,** $\frac{dH}{dr}$ versus $r$ at $\phi = 0.61$; $H = H_L + H_T$. Inset: comparison of $\xi_L$ with $\xi^{CM\perp}$ at discrete values of $r$ corresponding to the extrema of $\frac{dH}{dr}$.

extrema of $\frac{dH}{dr}$ (or extrema of $Z_{rel}$) wherein the net hydrodynamic correlations are weakest in direction orthogonal to the centre-of-mass displacements (Fig. 4b). In the inset to Fig. 4b, $\xi^{CM\perp}$ at extrema of $\frac{dH}{dr}$ are compared to the corresponding value of $\xi^L$ in the near-field. Evidently, thermal forces are the dominant fluctuations experienced by particles in the direction orthogonal to the centre-of-mass displacements, and thus the SER is recovered.

## Discussion

To conclude, we have measured and studied the near- and far-field longitudinal and transverse hydrodynamic modes in quasi-2D colloidal fluids. The findings highlight the importance of the contrasting magnitudes and phase-shift between these modes. These intrinsic features of 2D spatially confined systems lead to spatially inhomogeneous and anisotropic correlated dynamics of particles in colloid-pairs, and to the breakdown of the Stokes-Einstein relation (SER exponent $\xi < -1$). The microscopic insights gleaned suggest a mechanistic route to understand the unusual magnitude of $\xi$ observed here and in other studies[24,26]. Looking forward, these insights about near-field hydrodynamics of spherical particles could prove even more interesting for anisotropic particles, both passive and active, and may lead to novel ideas for affecting self-assembly and structural relaxation[12,31–33]. Broadly, we expect that these near-field hydrodynamics could impact phenomena in dense, spatially constrained systems, such as arise in cluster aggregation[34], translocation of proteins[15,16], nucleation and

growth kinetics of crystals[7,9], active systems[11,12], and clogging and jamming of channels[35,36]. Moreover, our experiments offer crucial insights about 2D hydrodynamics, especially in the near-field, which should stimulate development of simulation techniques that can account for hydrodynamics in dense suspensions of confined fluids at low Reynolds numbers[37,38].

## Methods
### Experimental details

We used polystyrene microspheres of diameter $\sigma = 1.04\,\mu m$ with size polydispersity of ~3% suspended in water. The particles were loaded into a wedge-shaped cell and allowed to sediment under gravity into the thin quasi-two-dimensional (quasi-2D) region of the cell (SI Figs. S4 and S5). Once a desired packing area-fraction, $\phi$, was achieved, the cell was equilibrated for at least six hours before performing video microscopy (SI Fig. S6). Data for all $\phi$s were taken from the same region of the cell. Experiments were performed at seven different particle packing area fractions in the range $0.15 \leq \phi \leq 0.61$ at temperature, $T = 22\,°C$. The typical number of colloids in the field-of-view at $\phi = 0.15$ were ~ 1100. We also checked that our choice of wedge-shaped cells for the experiments did not affect the dynamics measurements of the colloids in colloid-pairs (SI Fig. S5), unlike in Lancon et al.[39], which had much larger thickness variation. The images, at each $\phi$, were captured at 10 frames per second (fps) for 20 minutes. The trajectories of the particles were obtained using standard tracking algorithms[40]. The dynamic spatial resolution was found to be 20 nm. All subsequent analyses were performed using in-house developed codes.

### Displacement flow field

The hydrodynamic displacement field shown in 1f was determined as follows[5]. Briefly, we first computed the displacements of each particle, $\Delta \mathbf{r}_i(t = 0.5\,s)$. Next consider the pair of particles, $\{i, j\}$. $\Delta \mathbf{r}_i(t = 0.5\,s)$, $\Delta \mathbf{r}_j(t = 0.5\,s)$, and the unit vector ($\hat{\mathbf{r}}$) pointing from $i$ to $j$ subtend angles $\alpha_i$, $\alpha_j$, and $\beta_{ij}$ with respect to the positive $x -$ axis (ranging between 0 and $2\pi$). For each reference particle $i$ we rotate the 2D coordinate system through an angle $\alpha_i$ so that $\Delta \mathbf{r}_i(t = 0.5\,s)$ is aligned in the positive $x$ (horizontal) direction. The other vectors then also rotate through $\alpha_i$. We define the $\theta$ as the polar angle between the positive $x -$ axis (now aligned with $\Delta \mathbf{r}_i(t = 0.5\,s)$) and $\hat{\mathbf{r}}$ (SI Fig. S7). The polar plot $(r, \theta)$ in 1f is derived from the ensemble and initial time, $t_0$, average of the displacements.

### Dynamics measurement

Single particle diffusivity, $D^{self}$, in the lab frame at all values of $\phi$ was obtained from the measured mean squared displacements, $\langle \Delta r(t)^2 \rangle$ (SI Fig. S8). $\langle \Delta r(t)^2 \rangle = \langle \frac{1}{N} \sum_{k=1}^{N} (\Delta \mathbf{r}_k(t))^2 \rangle$. Here, $N$ is the total number of particles, $\Delta \mathbf{r}_k(t)$ is the displacement of $k^{th}$ particle during the lag time, $t$, and the averaging, $\langle \rangle$, were performed over $t_0$.

The diffusion of single particles, $D(r, \theta)$, with respect to the colloid-pair body-frame were measured from dynamical quantities, $\Delta r^2(t; \mathbf{r}', \mathbf{r} + \mathbf{r}', \theta)$, obtained using displacement of either of the particle $\{i, j\}$ in a pair along $\hat{\mathbf{R}}(\theta)\hat{\mathbf{r}}(t_0, \mathbf{r}', \mathbf{r} + \mathbf{r}')$); here $\hat{\mathbf{r}}$ is the unit vector along the line joining the particles in pair located at $\{\mathbf{r}', \mathbf{r} + \mathbf{r}'\}$ and at initial time $t_0$, and $\hat{\mathbf{R}}(\theta)$ is the rotation matrix. $D$ is obtained from the linear regime of $\langle \Delta r(t)^2 \rangle$ plot. Note, even $D(r)$ at low $\phi$, $\phi \leq 0.35$, are anisotropic when extracted from duration timescales $(t < 20\,s)$ where hydrodynamic interactions are significant (SI Fig. S9). At higher $\phi (\phi \geq 0.58)$, the dynamics become mildly sub-diffusive on short timescales, precluding extraction of $D(r, \theta)$, and hence, we cannot comment on whether $D(r)$ continues to be anisotropic at these densities at short timescales.

The structural relaxation time, $\tau_\alpha^{self}$, in the lab frame were measured from self-intermediate scattering functions, $F_s(\mathbf{q}, t)$[24] (SI Fig. S8). $F_s(\mathbf{q}, t) = \langle \frac{1}{N} \sum_{k=1}^{N} e^{i\mathbf{q} \cdot \Delta \mathbf{r}_k(t)} \rangle$, where symbols have usual meanings as explained above. For all the analyses presented in this study (including

pair dynamics), the magnitude of probe wave-vector, $q = 2\pi/a$, where $a$ is the position of the first peak in the pair correlation function, $g(r)$, at $\phi = 0.61$. The direction of $\mathbf{q}$ is chosen to be along $x - $axis.

The structural relaxation time, $\tau_\alpha(r, \theta)$, associated with particle motion with respect to the colloid-pair body-frame were measured using $F_s(\mathbf{q}_\theta, t; \mathbf{r}', \mathbf{r} + \mathbf{r}')$, by using $\Delta \mathbf{r}(t; \mathbf{r}', \mathbf{r} + \mathbf{r}', \theta)$ of either of the particle $\{i, j\}$ in a pair along $\hat{\mathbf{R}}(\theta)\hat{\mathbf{r}}(t_0, \mathbf{r}', \mathbf{r} + \mathbf{r}')$. $\mathbf{q}_\theta$ is along $\hat{\mathbf{R}}(\theta)\hat{\mathbf{r}}$. The time for which the decay of $F_s(q, t)$ drops to $1/e$ is read-off as structural relaxation time, $\tau_\alpha$, i.e., $F_s(q, t = \tau_\alpha) = 1/e$.

## Data availability
The microscopy videos and datasets generated during analyses in the current study are large ($>300$ GB) and hence are not publicly available, but can be made available from the corresponding author on request. Source data are provided with this paper.

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

## Acknowledgements

Authors thank Rajesh Ganapathy, Prasanna Venkatesh B., Adhip Agarwala, K. Hima Nagamanasa, Sankalp Nambiar, and Rajesh Singh for useful discussions. We gratefully acknowledge financial support from the Department of Science and Technology (Government of India), INSPIRE fellowship IF200274 (NHB), Indian Institute of Technology Gandhinagar, India (CKM) and the Start-up Research Grant of Science and Engineering Research Board of Government of India through SRG/2021/001077 (CKM), US National Science Foundation through Grant DMR2003659 (AGY) and the MRSEC Grant DMR1720530 including its optical microscopy shared experimental facility (AGY).

## Author contributions

N.H.B. and C.K.M designed the experiments and devised the experimental procedures. N.H.B performed the experiments and carried out the data analysis with inputs from A.G.Y and C.K.M. All authors contributed to the writing of the paper.

## Competing interests

The authors declare no competing interests.
