## [Peer Review File · Nature Communications]

REVIEWERS' COMMENTS

Reviewer #1 (Remarks to the Author):

The revision led to many clarifications and improvements of this work. I appreciate the new figures and discussions that the authors have included during the revision. However, I still do not think that this work solves an important problem that makes it interesting enough to be published in *Nature Physics*.

Some comments:

- In my previous report I mentioned that the paper would be more interesting if it presented equations for the actual near-field interactions. I understand that it is very difficult to derive such equations from first principles, but the authors could have constructed a fit function that is in line with their experimental results. Such a function would allow others to work with the authors' results. The figures and discussions of the authors show the reader that the interactions are complicated, but this does not give the reader some equation that he/she can use in own calculations or simulations (although this is one of the possible applications of this work mentioned by the authors).

- The breakdown of the Stokes-Einstein relation is still not surprising to me. Instead, this breakdown is to be expected in a situation as studied by the authors (the authors themselves mention in their rebuttal that the Stokes-Einstein relation is derived for conditions that are not fulfilled by the system addressed in this work so that one cannot expect the Stokes-Einstein relation to hold). It would have been an interesting result if the authors found that the Stokes-Einstein relation does hold. The papers on the Stokes-Einstein relation in 3D mentioned by the authors are interesting particularly since they show that the Stokes-Einstein relation holds in 3D.

- The authors did not convince me that deviations of the shape of their particles from a perfect sphere have no effect on the results.

Reviewer #2 (Remarks to the Author):

Upon evaluating the revised manuscript and the authors' response letter, it's clear the authors have addressed the initial comments with (minimal) revisions. The adjustments made, while not comprehensive enough for the standards of Nature Physics, do not detract from the inherent value of the experimental results concerning the hydrodynamics and correlated motion of colloids in quasi-2D confinement.

Given this, I believe that the manuscript is now suited for a publication in Nature Communications. In fact, the findings, albeit needing further refinement in future work, offer a baseline for continued discourse and exploration in the field. In particular, they might motivate the numerical and theoretical understanding suggested in the previous remarks to the authors.

Reviewer #1 (Remarks to the Author) and our Responses

Comments (general): The revision led to many clarifications and improvements of this work. I appreciate the new figures and discussions that the authors have included during the revision. However, I still do not think that this work solves an important problem that makes it interesting enough to be published in Nature Physics.

Response: We thank Reviewer #1 for the positive assessment of the clarifications and improvements we made in the revised manuscript.

Comments: Some comments:

In my previous report I mentioned that the paper would be more interesting if it presented equations for the actual near-field interactions. I understand that it is very difficult to derive such equations from first principles, but the authors could have constructed a fit function that is in line with their experimental results. Such a function would allow others to work with the authors' results. The figures and discussions of the authors show the reader that the interactions are complicated, but this does not give the reader some equation that he/she can use in own calculations or simulations (although this is one of the possible applications of this work mentioned by the authors).

Response: Thanks for these comments. Indeed, given the complexity of the hydrodynamic interactions in 2D, constructing precise equations for the near-field interactions from first principles is an extremely challenging task. We mentioned that the hydrodynamic correlations decay as $1/r^2$ in the far-field (**shown in Figure 1g, by solid and dashed lines**) and that, “*In addition, ..., experiments have shown that near-field drag correlations exhibit oscillatory modulations with respect to particle separation that are in-phase with structural signatures such as the particle pair correlation function [19].*” (Paragraph #1 of the Introduction). While one could readily fit a trigonometric/polynomial function to the observed data, we felt that such an endeavor, without consideration of its underlying physical significance, may not be helpful. Instead, it is desirable to construct a function that connects the underlying structure, e.g., the pair correlation function ($g(r)$), to the observed hydrodynamic correlations at different area fractions; this task, we believe, resides outside the scope of the current work. Nonetheless, our experiments reveal valuable insights, especially the observed phase lag between the longitudinal drag and transverse anti-drag hydrodynamic correlations and their linear combinations in the near-field interactions, which could be crucial for developing and testing such a theory.

Comments: The breakdown of the Stokes-Einstein relation is still not surprising to me. Instead, this breakdown is to be expected in a situation as studied by the authors (the authors themselves mention in their rebuttal that the Stokes-Einstein relation is derived for conditions that are not fulfilled by the system addressed in this work so that one cannot expect the Stokes-Einstein relation to hold). It would have been an interesting result if the authors found that the Stokes-Einstein relation does hold. The papers on the Stokes-Einstein relation in 3D mentioned by the authors are interesting particularly since they show that the Stokes-Einstein relation holds in 3D.

Response: Thanks for the remarks. Strictly speaking, the Stokes-Einstein relation (SER) is derived only in the asymptotic (far-field) limit and for dilute suspensions. However, it has been observed to hold true in many studies of dense suspensions in 3D. Our work is the first study to investigate the particle-pair separation (r –dependent) validity of the SER in quasi-2D. We show that the hydrodynamic correlations in quasi-2D (distinct from 3D) lead to the violation of SER in both the near-field (*which was not known*) and the far-field (*which is unexpected*). The identification of mechanistic routes leading to the breakdown of the

SER in 2D systems enables us to devise schemes for its recovery. Note, we had highlighted the practical significance of the SER relation in our previous response and manuscript.

Changes to the main manuscript:

Modification towards the end of paragraph #2 of the Introduction: Interestingly, although originally derived for dilute suspensions, the SER has been found to be valid in dilute and dense suspensions in three and higher spatial dimensions [23]. However, the microscopic origins of its breakdown in the reduced dimensions remains unclear [23–26]. Our experiments reveal unique features of 2D hydrodynamic interactions that lead to violation of the SER in confined geometries and identify scenarios that recover the SER. This work takes steps towards elucidating the microscopic origins of SER breakdown in reduced dimensions which to date have been unclear [23–26].

Comments: The authors did not convince me that deviations of the shape of their particles from a perfect sphere have no effect on the results.

Response: Thanks for your comments. The colloids used in our experiments were commercially procured and are known to have smooth and spherical shapes. In fact, the typical surface roughness even for large colloidal particles ($\sim 100 \mu m$), where roughness and sphericity become challenging to optimize in chemical reactions, falls within the range of a few nanometers only [Langmuir, **24**, 14, 7528–7531 (2008)]. However, to further convince the Reviewer, we have done the following analysis.

1. We have measured the apparent eccentricities, ($e^{apparent}$), of all the particles in the field-of-view at the largest area fraction ($\phi = 0.61$). Any deviation from the spherical shape will be reflected in a broader distribution of $e^{apparent}$. The distribution of $e^{apparent}$ is shown below.

The mean apparent eccentricity, $\langle e^{apparent} \rangle = 0.06$. This translates to an ~ 3 nm difference between the major and minor axis of the particles; the shape of the colloids is spherical.

2. In addition, to show that this tiny difference between the apparent major and minor axis of the particles has no bearing on the measured dynamics, we computed mean squared displacements, $\langle \Delta r^2(t) \rangle$ and $H^{L,T}(r, t)$ of particles in two different bins based on their eccentricities: $e^{apparent} \leq \langle e^{apparent} \rangle$ and $e^{apparent} > \langle e^{apparent} \rangle$ (see Figure below). Once again, these new analyses

reinforce our conclusion that shape of the particles in our study are spherical and that the minuscule variation of shape of the particles has no influence on the inferences drawn in our study.

Reviewer #2 (Remarks to the Author) and our Responses

Comments (general): Upon evaluating the revised manuscript and the authors' response letter, it's clear the authors have addressed the initial comments with (minimal) revisions. The adjustments made, while not comprehensive enough for the standards of *Nature Physics*, do not detract from the inherent value of the experimental results concerning the hydrodynamics and correlated motion of colloids in quasi-2D confinement.

Response: We thank Reviewer #2 for appreciating the standalone value of our experimental work.

Comments (general): Given this, I believe that the manuscript is now suited for a publication in *Nature Communications*. In fact, the findings, albeit needing further refinement in future work, offer a baseline for continued discourse and exploration in the field. In particular, they might motivate the numerical and theoretical understanding suggested in the previous remarks to the authors.

Response: We thank Reviewer #2 for finding our work suitable for publication in *Nature Communications*. We would like to add that we are also excited about our findings per unique features of hydrodynamic interaction in confined geometries, which has potential to be a benchmark for future theoretical/numerical investigations.